# The Effect of Wearing Elastic Compression Stockings on Leg Edema in Pregnant Women in Late Pregnancy as Determined by Measuring the Deep Venous Velocity and Flow

**DOI:** 10.3390/healthcare13030214

**Published:** 2025-01-21

**Authors:** Kanon Mori, Masafumi Koshiyama, Yumiko Watanabe, Noriko Okamoto, Nami Yanagisawa, Airi Banba, Eri Ikuta, Ayumi Ono, Keiko Seki, Miwa Nakagawa, Shin-ichi Sakamoto, Yoko Hara, Akira Nakashima

**Affiliations:** 1Department of Women’s Health, Graduate School of Human Nursing, The University of Shiga Prefecture, Hikone 522-0057, Japan; 2Graduate School of Human Nursing, The University of Shiga Prefecture, Hikone 522-0057, Japan; 3Faculty of Nursing, Shitennoji University, Habikino 583-0868, Japan; 4Nagahama City Kohoku Hospital, Nagahama 529-0493, Japan; 5Faculity of Advanced Engineering, The University of Shiga Prefecture, Hikone 522-0057, Japan; 6Iris Women’s Clinic, Alice, Hikone 522-0057, Japan

**Keywords:** maximum venous velocity, average venous velocity, venous blood flow, Doppler ultrasonography

## Abstract

**Background:** The present study evaluated the objective effectiveness of wearing compression stockings during late pregnancy on leg edema by measuring changes in venous velocity and blood flow. **Methods:** Using Doppler ultrasonography, we calculated the popliteal venous velocity and blood flow and compared them in the following four groups: (1) 13 non-pregnant women (26 legs) without leg edema, (2) 23 pregnant women (46 legs) in late pregnancy without leg edema, (3) 22 pregnant women (44 legs) in late pregnancy with leg edema who were followed up without treatment, and (4) 21 pregnant women (42 legs) in late pregnancy with leg edema who wore elastic compression stockings for 1 week. **Results:** Both the average velocity and blood flow of the popliteal vein of pregnant women’s legs in late pregnancy were significantly lower than those in non-pregnant women (6.32 ± 0.28 cm/s vs. 9.14 ± 0.37 cm/s; 118.48 ± 8.83 mL/min vs. 177.73 ± 11.74 mL/min, *p* < 0.0001, respectively). Furthermore, both the average venous velocity and blood flow in edematous legs in late pregnancy were significantly lower than those in non-edematous legs (5.32 ± 0.93 cm/s vs. 6.32 ± 0.28 cm/s; 82.68 ± 35.90 mL/min vs. 118.48 ± 8.83 mL/min, *p* < 0.0001, respectively). Finally, both the average venous velocity and blood flow were significantly higher in edematous legs in late pregnancy after wearing stockings than without treatment (7.69 ± 0.17 cm/s vs. 5.36 ± 0.17 cm/s; 143.22 ± 48.74 mL/min vs. 97.03 ± 40.07 mL/min, *p* < 0.0001, respectively). **Conclusions:** The wearing of elastic compression stockings by women with edematous legs in late pregnancy significantly increases the deep venous velocity and flow. Thus, it is possible to prevent deep vein thrombosis in this population.

## 1. Introduction

Leg edema during pregnancy is very common, occurring in about 80% of all pregnancies. It is normal when it is not associated with pre-eclampsia [1]. During normal pregnancy, complete body water increases by 6–8 L. Four–six L are extracellular, and no less than 2–3 L are interstitial [2]. Thus, the common leg edema during pregnancy is physiological edema. Hormonal changes cause changes in vascular permeability, promoting edema [2]. Furthermore, an enlarged uterus presses on the inferior vena cava. Thus, increased plasma volume and capillary permeability are caused by venous hypertension of the lower extremities [3,4].

Edema results from alterations in the basic physiological mechanisms governing fluid balance [5]. The hydrostatic pressure within the vascular system and colloid oncotic pressure in the interstitial fluid advance the movement of fluid from the vasculature to the extravascular space. Reduced plasma colloid oncotic pressure and obstruction of lymphatic flow in late pregnancy weaken the reabsorption of fluid into the intravascular space [6].

Treatments used for leg edema include leg elevation, water immersion, massage, intermittent pneumatic compression, use of medication, reflexology, bandages, and elastic stockings [7]. Of these treatments, the most common treatments for leg edema during pregnancy are leg elevation and compression stockings. We previously reported that wearing elastic compression stockings on the lower legs improves lower leg edema in pregnant women [7]. Using ultrasonography, we measured the thickness of the lower leg skin before and after wearing stockings. While the thickness of the lower leg skin increased in treatment-free pregnant women with leg edema, it significantly decreased in pregnant women with leg edema wearing elastic stockings [7]. At that time, we could not detect the movement of water flow.

However, the incidence of venous thromboembolism (VTE) in pregnant women is reportedly high. The frequency of VTE was 7.5 per 10,000 pregnancies in the Japanese population registered between 2011 and 2014 [8], but 3.26 per 10,000 person-years in Japanese users of low-dose estrogen progestin (LEP) between 2009 and 2013 (the former was 2.3 times the latter) [9]. In brief, pregnancy is a prothrombotic state. Pregnant women have all components of Virchow’s triad: hypercoagulability, endothelial damage, and venous stasis [10].

During pregnancy, a hypercoagulable state is initiated. Fibrin generation and levels of coagulation factors II, VII, VIII, and X are increased. But fibrinolytic activity is decreased [11,12].

Endothelial damage in the pelvic and leg veins can occur due to venous hypertension or during delivery [11]. Pelvic vein thrombosis accounts for 6–11% of cases of deep vein thrombosis (DVT) during pregnancy and the puerperium [13].

Venous stasis results from a decrease in venous tone and obstruction of venous flow by an enlarging uterus. However, little is known about its degree in pregnancy and the relationship between its degree and edema formation.

In the present study, we investigated the deep venous velocity and flow in non-pregnant women as well as women in late pregnancy using portable Doppler ultrasonography. Furthermore, we measured those values in the edematous legs and calculated them before and after wearing stockings. In brief, we studied how venous velocity and blood flow change during pregnancy, under conditions of leg edema, and while wearing elastic compression stockings. We made a hypothesis that venous stasis might worsen during pregnancy, under conditions of leg edema, but improve by wearing elastic stockings. We thus examined the possibility of preventing DVT in late pregnancy using elastic compression stockings.

## 2. Materials and Methods

### 2.1. Study Subjects and Study Approval

The controls were 18 participants (36 legs): non-pregnant women, students, and female teaching staff at our university. Thirteen (26 legs) control women had no leg edema, and five (10 legs) had leg edema. We further investigate 66 research subjects: pregnant women (132 legs) who attended an outpatient clinic in the department of Obstetrics and Gynecology, Hikone City, Shiga, Japan, at 36 weeks of gestation. Research subjects at 36 weeks of gestation were 23 pregnant women (46 legs) without leg edema, 22 pregnant women with leg edema (44 legs) who did not wear stockings for 1 week, and 21 pregnant women (42 legs) with leg edema who wore stockings for 1 week.

The study protocol was approved by the Ethics Committee of the University of Shiga Prefecture (No. 917; 13 April 2023). All participants provided their written informed consent prior to study entry.

### 2.2. Assessment of the Grade of Pitting Grading

Pregnant women at 36 gestational weeks were diagnosed with lower leg edema using the pitting edema. Finger pressure was applied to the swollen area of the skin of the anterior surface of the tibia to determine whether an indentation formed that persisted after the removal of pressure. The women wore elastic compression stockings from 36 to 37 weeks of gestation. Participants were asked not to alter their lifestyle during the study period (they did not change much in activity, diet, and hydration).

We used the pitting edema method to determine the grade of skin edema on the lower leg. The grades of pitting edema were as follows: Grade 0, negative for edema; Grade 1, mild pitting edema that disappeared within 10 s; Grade 2, moderate pitting edema that disappeared after 10–15 s; Grade 3, severe pitting edema that lasted for more than 15 s [14].

### 2.3. Measurements of the Deep Venous Velocity and Flow

To investigate the changes in venous stasis, we divided the participants into the following groups: (1) non-pregnant women without leg edema, (2) pregnant women without leg edema at 36 gestational weeks, (3) pregnant women with leg edema at 36 gestational weeks who were followed up without treatment, and (4) pregnant women with leg edema who wore elastic compression stockings from 36 to 37 gestational weeks. We measured the deep venous velocity and flow in these participants. This study was conducted every morning.

Measurements of the deep (popliteal) vein were performed using portable Doppler ultrasonography. The experiments were performed using an Arietta Prologue (Fujifilm-Healthcare, Tokyo, Japan) with a linear probe (10 MHz) (Figure 1A). As a fixed point, the center of the probe was set vertically on the popliteal fossa (Figure 1B). The participants remained calm for 10 min before measurements, and they sat with their legs stretched out.

A Doppler examination of the popliteal vein was performed on the left and right sides of each subject. The angle of the sampling gate did not exceed 60°. At least three waveforms were described (Figure 2A). First, the horizontal bar was aligned with the top of the second tall wave to measure the maximum venous velocity (cm/s) (Figure 2A). Next, the vertical bar was aligned with both ends of the second tall wave to measure the average venous velocity (cm/s) (Figure 2B). Subsequently, the popliteal vein diameter (mm) and cross-sectional area (cm^2^) were measured to calculate the venous blood flow (mL/min). The same trained evaluator (who had been trained for three months) assessed the measured values under the appropriate conditions.

The elastic compression stockings (Akiyama Co., Tokyo, Japan), which have medical applications, were worn on the edema-positive legs of the pregnant women. The compressive pressures of below-knee graduated elastic compression stockings were 27 mmHg on the ankle and 18 mmHg on the calf. The pregnant women wore elastic compression stockings from 36 to 37 weeks of gestation every day except for bathing and sleeping.

We measured the maximum and average venous velocities and the venous blood flow in the four (1–4) groups. The latter two groups (3 and 4) were measured twice: at 36 and 37 gestational weeks. Particularly noteworthy are the differences in venous stasis among the four study groups.

### 2.4. Statistical Analyses

The study design was a non-randomized, controlled intervention trial. Statistical analyses were performed using the JMP Pro statistical version 14 (SAS Institute Japan, Tokyo, Japan) and GraphPad Prism, version 9 (GraphPad Software, San Diego, CA, USA) software programs. The results of the maximum and average venous velocities and the venous blood flow are presented as the mean ± SEM (mm) and were compared between non-pregnant women without edema and pregnant women without edema at 36 gestational weeks using Student’s *t*-test. They were further compared among grades 0, 1, and 2 of edema using a one-way analysis of variance (ANOVA). Comparisons among the maximum and average venous velocities and venous blood flow in the pregnant women were performed using Student’s *t*-test and paired *t*-test.

We calculated the effect size of this study (minimum sample size) using an α level of 5% and a power of 90%. The required sample size was calculated based on a comparison between the mean ± standard error of the mean (SEM) measurements of the maximum venous velocity, average venous velocity, and venous blood flow of the 10 edematous legs of 5 pregnant women without treatment from 36 to 37 weeks and those of the 10 edematous legs of 5 pregnant women with elastic stockings from 36 to 37 weeks (α level, 5%; power, 90%). Thus, the required sample size was calculated as 21 women (42 legs). Statistical significance was set at *p* < 0.05.

## 3. Results

### 3.1. Clinical Characteristics of the Study Participants (Table 1)

The clinical characteristics of the participants are shown in Table 1. The mean age of the pregnant women, ± SEM was 31.12 ± 4.76 (range, 19–41) years old, while the mean age of the control women ± SEM was 29.78 ± 7.95 (range, 21–44) years old (*p* = 0.2). In addition, the mean ages ± SEM of the three types of pregnant women at 36 gestational weeks (non-edematous, edematous, edematous with stocking) were 30.30 ± 3.87, 31.04 ± 6.62, and 32.05 ± 2.84 years old, respectively. All non-pregnant and pregnant women did not have serious complications. Edematous legs were not pre-eclampsia but physiological.
healthcare-13-00214-t001_Table 1Table 1Clinical characteristics of the study participants.ControlsSubjects*p* Value18 non-pregnant women: 36 legs66 pregnant women: 132 legs
**Age**29.78 ± 7.98 y31.12 ± 4.76 y0.226 non-edematous

and 10 edematous legs46 non-edematous legs vs. 44 edematous legs vs. 42 edematous legs0.16**Age**30.30 ± 3.87 y31.04 ± 6.62 y32.05 ± 2.84 y0.12**Primipara**24 legs24 legs16 legs
**Multipara**22 legs20 legs26 legs0.19


### 3.2. A Comparison of the Maximum and Average Venous Velocities as Well as the Venous Blood Flow Between Non-Pregnant Women and Pregnant Women in Late Pregnancy

First, we investigated how maximum and average venous velocities or venous blood flow change during pregnancy. We compared the maximum venous velocity of the popliteal vein in the legs of 26 non-pregnant women and that of 46 pregnant women at 36 gestational weeks. All participants had legs without edema (Figure 3A), with the maximum velocity of the popliteal vein being significantly lower in the legs of pregnant women than in non-pregnant women (6.88 ± 0.31 cm/s vs. 10.17 ± 0.42 cm/s; 32.4% (10.17–6.88/10.17) decrease, *p* < 0.0001). The average venous velocity of pregnant women at 36 gestational weeks was also significantly lower than that of non-pregnant women (6.32 ± 0.28 cm/s vs. 9.14 ± 0.37 cm/s; 30.9% decrease, *p* < 0.0001, Figure 3B), as was the venous blood flow of pregnant women at 36 gestational weeks compared with that of non-pregnant women (118.48 ± 8.83 mL/min vs. 177.73 ± 11.74 mL/min; 33.3% decrease, *p* < 0.0001, Figure 3C).

### 3.3. A Comparison of the Maximum and Average Venous Velocities, and the Venous Blood Flow Among Degrees of Leg Edema in Late Pregnancy

Next, we investigated the level changes in the maximum venous velocity in pregnant women at 36 gestational weeks, with and without leg edema. The maximum venous velocity of pregnant women at 36 weeks with grade 1 and 2 edema was significantly lower than that of those women with grade 0 edema (no edema) (5.64 ± 0.95 cm/s and 5.77 ± 1.28 cm/s vs. 6.88 ± 0.31 cm/s, 28.0% decrease and 16.1% decrease, *p* < 0.0001, respectively, Figure 4A). The average venous velocity of grade 1 and 2 edema was also significantly lower than that of those women with grade 0 edema (no edema) (5.32 ± 0.93 cm/s and 5.35 ± 1.36 cm/s vs. 6.32 ± 0.28 cm/s, 15.8% decrease and 15.3% decrease, *p* < 0.0001 and *p* = 0.001) (Figure 4B). In addition, the venous blood flow of pregnant women with grade 1 and 2 edema was also significantly lower than that of those with grade 0 edema (no edema) (82.68 ± 35.90 mL/min and 71.95 ± 29.90 mL/min vs. 118.48 ± 8.83 mL/min, 30.2% decrease and 39.3% decrease, *p* < 0.0001, respectively, Figure 4C).

### 3.4. Level Changes in the Maximum and Average Venous Velocities and the Venous Blood Flow in the Edematous Leg in Late Pregnancy Before and After Wearing Elastic Compression Stockings

Finally, we investigated the influence of wearing elastic compression stockings for edematous legs during late pregnancy on the maximum and average venous velocities and the flow volume. Twenty-one pregnant women with 42 edematous legs wore elastic compression stockings from 36 to 37 gestational weeks, while 22 pregnant women with 44 edematous legs did not wear stockings during the study period. There was no significant difference in the maximum venous velocity at 36 gestational weeks between the two groups (stockingless group vs. before wearing stockings group) (5.44 ± 0.15 cm/s vs. 5.94 ± 0.16 cm/s, *p* = 0.05, Figure 5(A1a)). However, the maximum venous velocity was significantly higher in the edematous legs of pregnant women after wearing stockings than in pregnant women without stockings (8.60 ± 1.53 cm/s vs. 5.85 ± 1.04 cm/s, 47.0% increase, *p* < 0.0001, Figure 5(A1b)). The level changes in maximum venous velocity in pregnant women with elastic stockings were significantly higher than those in pregnant women without stockings (2.66 ± 1.39 cm/s vs. 0.41 ± 1.13 cm/s, paired *t*-test: *p* < 0.0001 vs. *p* = 0.02, Figure 5(A2a,A2b)).

Similarly, there was no significant difference in the average venous velocity at 36 gestational weeks between the two groups (stockingless group vs. before wearing stockings group) (5.10 ± 0.94 cm/s vs. 5.58 ± 1.13 cm/s, *p* = 0.05, Figure 5(B1a)). However, the average venous velocity was also significantly higher in the edematous legs of pregnant women after wearing stockings than in pregnant women without stockings (7.69 ± 0.17 cm/s vs. 5.36 ± 0.17 cm/s, 43.5% increase, *p* < 0.0001, Figure 5(B1b)). The level changes in average venous velocity in pregnant women with elastic stockings were significantly higher than those in pregnant women without stockings (2.11 ± 1.45 cm/s vs. 0.25 ± 1.04 cm/s, paired *t*-test: *p* < 0.0001 vs. *p* = 0.10, Figure 5(B2a,B2b)).

The same trend was observed for the venous blood flow. There was no significant difference in the venous blood flow at 36 gestational weeks between the two groups (stockingless group vs. before wearing stockings group) (84.29 ± 39.94 mL/min vs. 77.27 ± 30.01 mL/min, *p* = 0.36, Figure 5(C1a)). However, the venous blood flow was also significantly higher in the edematous legs of pregnant women after wearing stockings than in pregnant women without stockings (143.22 ± 48.74 mL/min vs. 97.03 ± 40.07 mL/min, 47.6% increase, *p* < 0.0001, Figure 5(C1b)). The level changes in the venous blood flow in pregnant women with elastic stockings were significantly higher than those in pregnant women without stockings (65.95 ± 48.78 mL/min vs. 12.74 ± 36.72 mL/min, paired *t*-test: *p* < 0.0001 vs. *p* = 0.06, Figure 5(C2a,C2b)).

## 4. Discussion

In the present study, the maximum and average venous velocities of pregnant women at 36 gestational weeks were significantly lower in the popliteal vein than in non-pregnant women (32.4% and 30.9% decreases, respectively). These results are consistent with those of a previous report. Palmgren et al. reported that the maximal peaks and mean velocities in the popliteal vein in pregnant women were significantly lower than those in non-pregnant women during the last trimester of pregnancy using color Doppler equipment [15]. Owing to uterine enlargement during pregnancy, a decrease in the venous blood velocity and an increase in the venous blood pressure in the lower limbs may occur. Baumann et al. reported that reduction in venous femoral blood flow and an increase in the femoral vein diameter during pregnancy might be associated with the common occurrence of venous disorders [16].

Recently, Jensen et al. reported that slowing the venous flow in the lower extremities on Doppler ultrasound increased the rate of subsequent DVT development in 975 oncology patients [17]. Interestingly, in cases of subsequent DVT development where a baseline bilateral examination of Doppler ultrasound was performed (N = 6) and unilateral DVT developed (N = 5), it most often developed in the leg with a relatively slow flow. This finding indicates that venous stasis is a risk factor for DVT. In response to this fact, we can explain why the rate of DVT formation in late pregnancy is higher than in non-pregnancy.

The maximum and average venous velocities of pregnant women at 36 gestational weeks with grade 1 and 2 leg edema were significantly lower than those of non-edema pregnant women at 36 weeks (28.0% and 16.1% decreases; 15.8% and 15.3% decreases, respectively). In addition, the venous blood flows of pregnant women with leg edema of grades 1 and 2 were also significantly lower than those of pregnant women without edema (30.2% and 39.3% decreases, respectively). These results are new findings and indicate that the risk of DVT formation was higher in pregnant women with leg edema during late pregnancy than in those without leg edema.

A decrease in the venous blood flow results from the movement of water. Water moves from the venous intravascular space to the extravascular tissues. This phenomenon is likely to reduce the venous velocity and blood flow volume. Previously, we proved that the subcutaneous skin thickness of legs with edema in late pregnancy is increased using ultrasound sonography [7,14]. The present results support our previous findings and provide new insights. The prevalence of DVT was reportedly significantly higher in people with positive leg edema than in those with negative leg edema after the 2016 Kumamoto Earthquake in Japan, and leg edema is a risk factor for DVT formation [18].

We previously reported the effect of wearing elastic compression stockings as a treatment for leg edema in late pregnancy, finding that the subcutaneous skin thickness of the legs significantly decreased [14]. It is assumed that water moves from the extravascular space into the intravascular space by wearing these stockings. This phenomenon supports the facts that the maximum and average venous velocities and flow volumes were increased by wearing stockings.

Cimunova et al. demonstrated that wearing test footwear (slippers and sneakers or winter shoes) might prevent popliteal venous blood velocity reduction during advanced phases of pregnancy [19]. They investigated the experimental effect in no-edematous pregnant women in advanced phases of pregnancy. For edematous legs, wearing elastic compression stockings makes sense and results in a significant increase in the average venous velocity (43.5% increase) and venous flow (47.6% increase). Some treatments used for leg edema include leg elevation, water immersion, massage, intermittent pneumatic compression, medication use, reflexology, and bandages, as well as elastic stockings [20,21]. Of these, one of the most common treatments for leg edema during pregnancy is compression stocking use [22]. It is easy to use. Allegra et al. reported that leg symptoms and pain were reduced in pregnant women who wore compression stockings [23]. They also demonstrated the importance of wearing compression stockings continuously to improve the quality of life among pregnant women. Saliba Junior et al. found that there was reflux in the diameters of the great saphenous vein (GSM) and the small saphenous vein in 0 of 30 pregnant women in the stocking-use group and in 16 of 30 in the group that did not wear the stockings using Doppler ultrasonography [24]. These phenomena indicate that wearing compression stockings can prevent venous reflux and hypertension [25,26].

In the present study, the risk of DVT formation was higher in the legs of women in late pregnancy than in those of non-pregnant women. Actually, it was reported that the incidence of pregnancy-related venous thromboembolism was 1 in 1500 deliveries, and the risk of venous thromboembolism was five times higher in a pregnant woman than in a non-pregnant woman [27]. In addition, the risk of DVT formation is also higher in edematous legs of women in late pregnancy than in non-edematous legs of this population. In the present study, the venous velocity and blood flow were significantly increased by wearing elastic compression stockings for leg edema in women in late pregnancy, and these phenomena are new discoveries. Thus, pregnant women in late pregnancy with edematous legs should wear elastic compression stockings to prevent DVT, which is the application of this study. By wearing these stockings, water moves from extravascular spaces to venous intravascular space, and this phenomenon can prevent DVT formation. Several other authors insist on the same [24,28]. In addition, the effect of compression stockings on maternal and fetal circulation was studied in the left lateral position and standing in 21 patients with uterovascular syndrome in late pregnancy [29]. Thus, the use of compression stockings indicated a measurable improvement in maternal and fetal circulation.

During pregnancy, a hypercoagulable state, endothelial damage and venous stasis have been reported as causes of DVT. We showed the effect of wearing stockings on venous stasis of leg edema in late pregnancy. However, we could not show their effect on venous stasis in all phases of pregnancy with and without leg edema. This is a limitation of this study. In the near future, we should show the effects of wearing elastic compression stockings to prevent DVT in larger, more diverse populations of pregnant women at all phases, by calculating the deep venous velocity and blood flow and observing DVT formation.

## 5. Conclusions

The risk of DVT formation is higher in the leg of women in late pregnancy than in those of non-pregnant women. In addition, the risk of DVT formation in edematous legs of women in late pregnancy is even higher than in non-edematous legs of women in late pregnancy. However, the venous velocity and blood flow were significantly increased by wearing elastic compression stockings for leg edema in women in late pregnancy. Thus, wearing stockings has the potential to reduce the risk of DVT in edematous legs during late pregnancy, though further studies are needed to confirm this effect across all pregnancy phases.

## Figures and Tables

**Figure 1 healthcare-13-00214-f001:**
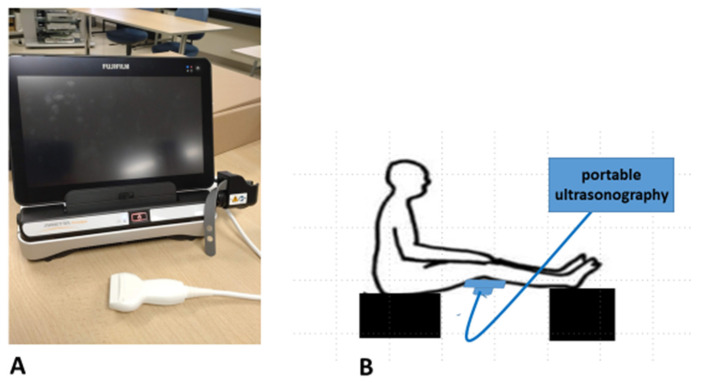
Measurements of the deep (popliteal) vein were performed using portable Doppler ultrasonography. (**A**) B-scan portable ultrasound device, Arietta Prologue (Fujifilm-Healthcare, Tokyo, Japan) with a linear probe (10 MHz). (**B**) The center of the probe was set vertically on the popliteal fossa.

**Figure 2 healthcare-13-00214-f002:**
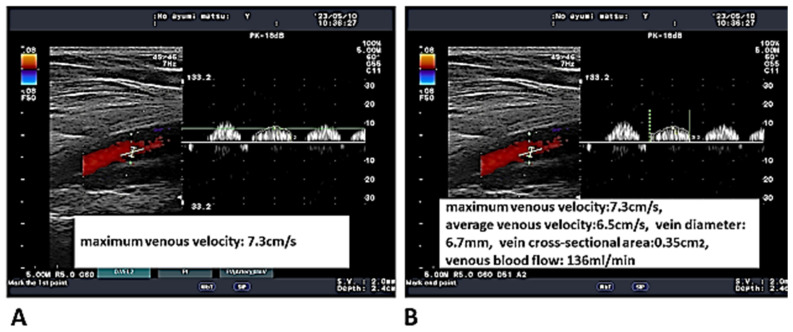
A Doppler examination of the popliteal vein was performed. (**A**) At least three waveforms were described. The horizontal bar was aligned with the top of the second tall wave to measure maximum venous velocity (cm/s). (**B**) The vertical bar was aligned with both ends of the second tall wave to measure average venous velocity (cm/s). The popliteal vein diameter (mm) and cross-sectional area (cm^2^) were measured to calculate the venous blood flow (mL/min).

**Figure 3 healthcare-13-00214-f003:**
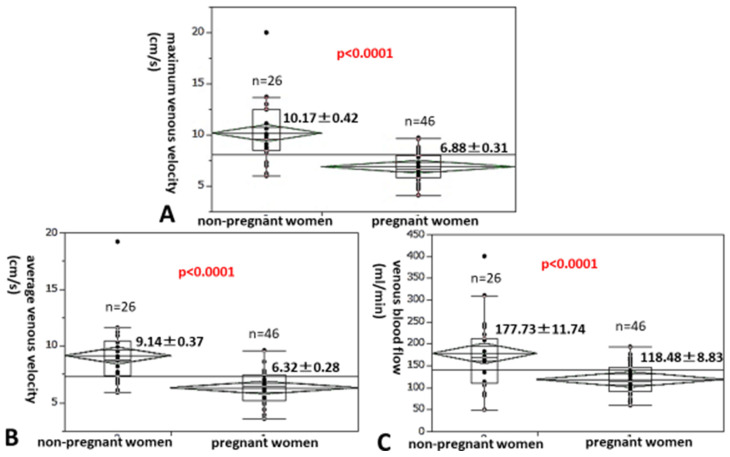
We compared the maximum (**A**) and average (**B**) venous velocities and the venous blood flow (**C**) of the popliteal vein in the legs of non-pregnant women with those of pregnant women at 36 gestational weeks.

**Figure 4 healthcare-13-00214-f004:**
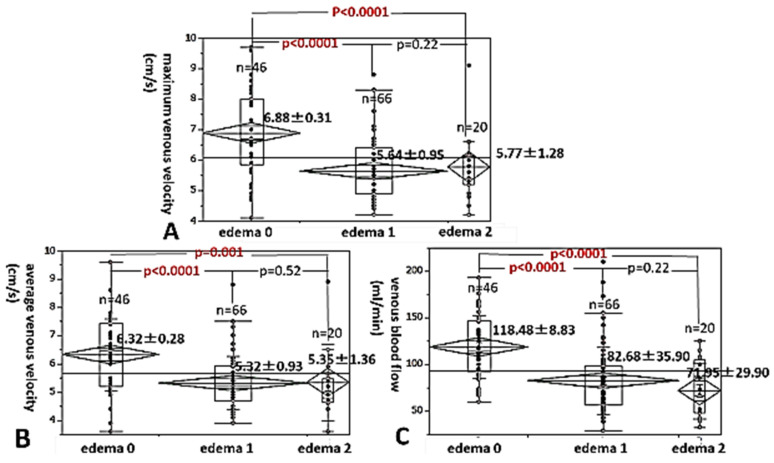
We investigated the level changes in the maximum (**A**) and average (**B**) venous velocities and the venous blood flow (**C**) in pregnant women at 36 gestational weeks, with and without leg edema.

**Figure 5 healthcare-13-00214-f005:**
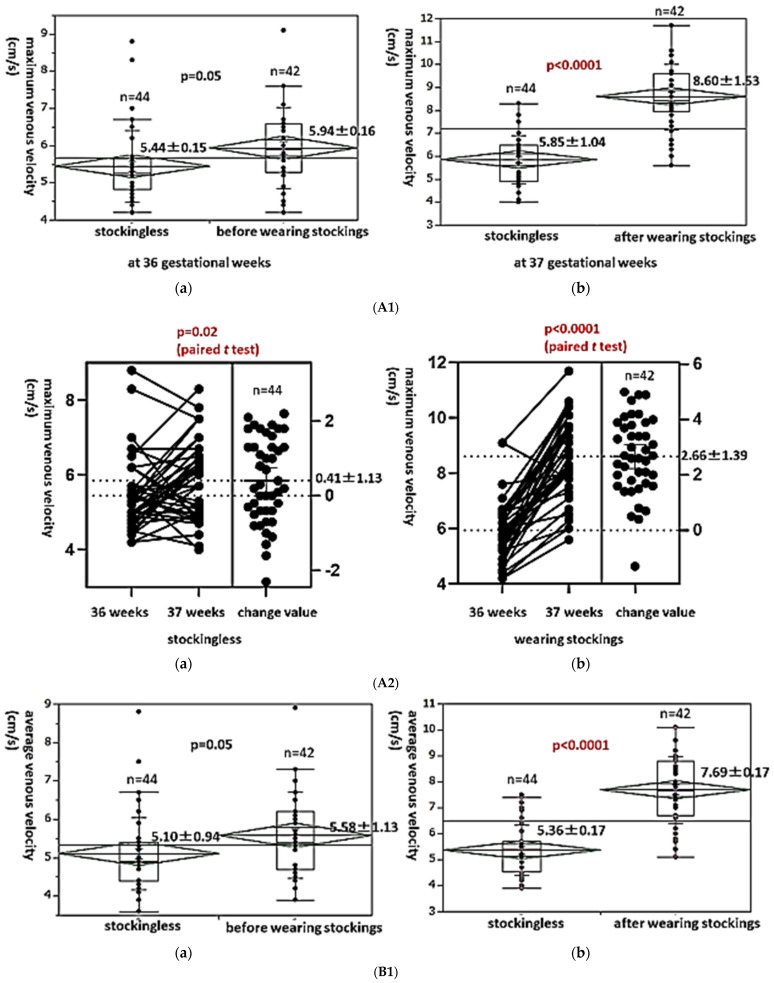
(**A1**) We investigated the influence of wearing elastic compression stockings for edematous legs during late pregnancy on the deep venous velocity and flow volume. There was no significant difference in the maximum venous velocity at 36 gestational weeks between the 2 groups (stockingless group vs. before wearing stockings group) (**a**). However, the maximum venous velocity was significantly higher in the edematous legs of pregnant women after wearing stockings than in pregnant women without stockings (**b**). (**A2**) The level changes in maximum venous velocity in pregnant women with elastic stockings were significantly higher than that in the pregnant women without stockings (**a**,**b**). (**B1**) There was no significant difference in the average venous velocity at 36 gestational weeks between the 2 groups (stockingless group vs. before wearing stockings group) (**a**). However, the average venous velocity was also significantly higher in the edematous legs of pregnant women after wearing stockings than in pregnant women without stockings (**b**). (**B2**) The level changes in average venous velocity in pregnant women with elastic stockings were significantly higher than those in pregnant women without stockings (**a**,**b**). (**C1**) There was no significant difference in the venous blood flow at 36 gestational weeks between the 2 groups (stockingless group vs. before wearing stockings group) (**a**). However, the venous blood flow was also significantly higher in the edematous legs of pregnant women after wearing stockings than in pregnant women without stockings (**b**). (**C2**) The level changes in the venous blood flow in pregnant women with elastic stockings were significantly higher than those in pregnant women without stockings (**a**,**b**).

## Data Availability

No new data were created or analyzed in this study. Data sharing is not applicable to this article.

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
