# Peer review of "The Effect of Wearing Elastic Compression Stockings on Leg Edema in Pregnant Women in Late Pregnancy as Determined by Measuring the Deep Venous Velocity and Flow"

_healthcare, 2025, doi:10.3390/healthcare13030214_

Round 1

Reviewer 1 Report

Comments and Suggestions for Authors

1. The study title focuses on elastic compression stockings, leg edema, and the specific parameters measured (deep venous velocity and flow).

2. The abstract summarizes key elements of the study, including the objective, methodology, and significant findings. Some sentences are dense and could be simplified for better readability, especially for readers unfamiliar with the topic.

3. The introduction provides a detailed background on the prevalence and causes of leg edema during pregnancy, references past findings, and explains the physiological mechanisms leading to edema and related complications. However, Providing a brief overview of current gaps in research regarding compression stockings and their broader application in clinical settings would enhance the study's relevance.

4. The research design appears appropriate for addressing the study's objectives, but here are some specific observations regarding its suitability and potential areas for enhancement: While non-pregnant women serve as a control group, including pregnant women without edema across all phases of pregnancy, could strengthen comparisons. Lack of blinding could introduce bias, especially during subjective assessments like edema grading. Although participants were asked not to alter their lifestyle, there is no mention of monitoring compliance. Variations in activity, diet, or hydration might influence results. Need to be clear on this aspect.

5. The results are well-organized and supported with quantitative data, but there are areas where clarity and presentation could be improved. Here’s a detailed analysis: Consistent terminology (e.g., "maximum venous velocity" vs. "venous velocity") across figures, tables, and text would avoid confusion

6. Add a statement about the limitations of the study and the need for further research to validate findings in larger, more diverse populations.

7. Rephrase the conclusion to explicitly connect the results with clinical implications, e.g., "The significant increase in venous velocity and flow after compression stocking use suggests their potential to reduce the risk of DVT in late pregnancy, though further studies are needed to confirm this effect across all pregnancy phases."

Comments on the Quality of English Language
  • Simplify overly complex sentences for clarity.
  • Ensure consistent use of technical terms throughout the text.
  • Use active voice selectively to improve engagement.
  • Review transitions to make the document more cohesive.

Author Response

We corrected our article with red color (highlight).

<Answers to the referee 1>

  1. The study title focuses on elastic compression stockings, leg edema, and the specific parameters measured (deep venous velocity and flow).=> (A)Thank you.
  2. The abstract summarizes key elements of the study, including the objective, methodology, and significant findings. Some sentences are dense and could be simplified for better readability, especially for readers unfamiliar with the topic.=>(A) Thank you very much for your kind comments. Yes, we deleted the following words: 30.9% decrease, 33.3 % decrease,15.8% decrease, 30.2% decrease, 43.5 % increase, 47.6 % increase,
  3. The introduction provides a detailed background on the prevalence and causes of leg edema during pregnancy, references past findings, and explains the physiological mechanisms leading to edema and related complications. However, Providing a brief overview of current gaps in research regarding compression stockings and their broader application in clinical settings would enhance the study's relevance.=> (A) Yes, we inserted following sentences in introduction: Treatments used for leg edema include leg elevation, water immersion, massage, intermittent pneumatic compression, use of medication, reflexology, bandage and elastic stockings [7]. Of these treatments, the most common treatments for leg edema during pregnancy are leg elevation and compression stockings.
  4. The research design appears appropriate for addressing the study's objectives, but here are some specific observations regarding its suitability and potential areas for enhancement: While non-pregnant women serve as a control group, including pregnant women without edema across all phases of pregnancy, could strengthen comparisons. Lack of blinding could introduce bias, especially during subjective assessments like edema grading. Although participants were asked not to alter their lifestyle, there is no mention of monitoring compliance. Variations in activity, diet, or hydration might influence results. Need to be clear on this aspect.=>(A) Thank you for good comments. Yes we inserted following sentences: (They didn’t change much in activity, diet and hydration) in Materials and Methods.
  5. The results are well-organized and supported with quantitative data, but there are areas where clarity and presentation could be improved. Here’s a detailed analysis: Consistent terminology (e.g., "maximum venous velocity" vs. "venous velocity") across figures, tables, and text would avoid confusion=>(A) Thank you very much for your nice pointing outs. We changed to the following sentences in Results :First, we investigated how maximum and average venous velocities or venous blood flow change during pregnancy and Finally, we investigated the influence of wearing elastic compression stockings for edematous legs during late pregnancy on the maximum and average venous velocities and the flow volume.
  6. Add a statement about the limitations of the study and the need for further research to validate findings in larger, more diverse populations.=> (A) Yes, we added the following sentences in Discussion: However, we could not show their effect on venous stasis in all phases of pregnancy with and without leg edema. This is a limitation of this study. In the near future, we should show the effects of wearing elastic compression stockings to prevent DVT in larger, more diverse populations of pregnant women at all phase, by calculating the deep venous velocity and blood flow and observing DVT formation.
  7. Rephrase the conclusion to explicitly connect the results with clinical implications, e.g., "The significant increase in venous velocity and flow after compression stocking use suggests their potential to reduce the risk of DVT in late pregnancy, though further studies are needed to confirm this effect across all pregnancy phases."=>(A) Thank you for good advice. Yes, we inserted the following sentence in Conclusion: Thus, wearing stockings has potential to reduce the risk of DVT in edematous legs during late pregnancy, though further studies are needed to confirm this effect across all pregnancy phases.

We corrected our article with red color (highlight).

<Answers to the referee 1>

  1. The study title focuses on elastic compression stockings, leg edema, and the specific parameters measured (deep venous velocity and flow).=> (A)Thank you.
  2. The abstract summarizes key elements of the study, including the objective, methodology, and significant findings. Some sentences are dense and could be simplified for better readability, especially for readers unfamiliar with the topic.=>(A) Thank you very much for your kind comments. Yes, we deleted the following words: 30.9% decrease, 33.3 % decrease,15.8% decrease, 30.2% decrease, 43.5 % increase, 47.6 % increase,
  3. The introduction provides a detailed background on the prevalence and causes of leg edema during pregnancy, references past findings, and explains the physiological mechanisms leading to edema and related complications. However, Providing a brief overview of current gaps in research regarding compression stockings and their broader application in clinical settings would enhance the study's relevance.=> (A) Yes, we inserted following sentences in introduction: Treatments used for leg edema include leg elevation, water immersion, massage, intermittent pneumatic compression, use of medication, reflexology, bandage and elastic stockings [7]. Of these treatments, the most common treatments for leg edema during pregnancy are leg elevation and compression stockings.
  4. The research design appears appropriate for addressing the study's objectives, but here are some specific observations regarding its suitability and potential areas for enhancement: While non-pregnant women serve as a control group, including pregnant women without edema across all phases of pregnancy, could strengthen comparisons. Lack of blinding could introduce bias, especially during subjective assessments like edema grading. Although participants were asked not to alter their lifestyle, there is no mention of monitoring compliance. Variations in activity, diet, or hydration might influence results. Need to be clear on this aspect.=>(A) Thank you for good comments. Yes we inserted following sentences: (They didn’t change much in activity, diet and hydration) in Materials and Methods.
  5. The results are well-organized and supported with quantitative data, but there are areas where clarity and presentation could be improved. Here’s a detailed analysis: Consistent terminology (e.g., "maximum venous velocity" vs. "venous velocity") across figures, tables, and text would avoid confusion=>(A) Thank you very much for your nice pointing outs. We changed to the following sentences in Results :First, we investigated how maximum and average venous velocities or venous blood flow change during pregnancy and Finally, we investigated the influence of wearing elastic compression stockings for edematous legs during late pregnancy on the maximum and average venous velocities and the flow volume.
  6. Add a statement about the limitations of the study and the need for further research to validate findings in larger, more diverse populations.=> (A) Yes, we added the following sentences in Discussion: However, we could not show their effect on venous stasis in all phases of pregnancy with and without leg edema. This is a limitation of this study. In the near future, we should show the effects of wearing elastic compression stockings to prevent DVT in larger, more diverse populations of pregnant women at all phase, by calculating the deep venous velocity and blood flow and observing DVT formation.
  7. Rephrase the conclusion to explicitly connect the results with clinical implications, e.g., "The significant increase in venous velocity and flow after compression stocking use suggests their potential to reduce the risk of DVT in late pregnancy, though further studies are needed to confirm this effect across all pregnancy phases."=>(A) Thank you for good advice. Yes, we inserted the following sentence in Conclusion: Thus, wearing stockings has potential to reduce the risk of DVT in edematous legs during late pregnancy, though further studies are needed to confirm this effect across all pregnancy phases.

Reviewer 2 Report

Comments and Suggestions for Authors

It is unclear what a gap of knowledge is in the study. The research design was inappropriate, and the number of samples was questionable. I did not understand why you examined four populations: sample size calculation and inclusion and exclusion criteria in the control group. 

Who measures or uses a sampling technique? Please also provide the accuracy, validity, and reliability of the measurement and record it. 

Results: please provide the data of the measurement in Table and also provide the statistical analysis.

Discussion: what is the new knowledge of the study? what is the application of the study. 

Author Response

We corrected our article with red color (highlight).

<Answers to the referee 2>

*It is unclear what a gap of knowledge is in the study. The research design was inappropriate, and the number of samples was questionable.==>(A) Thank you for your advice. We inserted the following sentence in Materials and Methods: The study design was non-randomized controlled and intervention trial. We calculated the effect size of the study (minimum sample size)using an α level of 5% and power of 90%. The required sample size was calculated based on a comparison between the mean ± SEM measurements of the maximum venous velocity, average venous velocity and venous blood flow of the 10 edematous legs in 5 pregnant women without treatment from 36 to 37 weeks and those of the 10 edematous legs in 5 pregnant women with elastic stockings from 36 to 37 weeks (α level, 5%; power, 90%), using statistical software. This is minimum sample. This is necessary and sufficient sample size because all data showed significant differences.

*I did not understand why you examined four populations: sample size calculation and inclusion and exclusion criteria in the control group. =>(A) Thank you for your advice. We firstly wanted to show the effect of elastic compression stockings and the fact the venous velocity and blood flow are significantly increased by wearing elastic compression stockings for leg edema in women in late pregnancy. From the perspective of DVT prevention, however, we thought that we had to calculate these indices in non-pregnant women, pregnant women with or without leg edema. It turned out that pregnant women with leg edema is most dangerous, followed by pregnant women without leg edema. Thus, we had to examine four population from the perspective of DVT prevention. However, we could not show their effect on venous stasis in all phases of pregnancy with and without leg edema. This is a limitation of this study (in Text). We inserted the following sentence in 2.4. Measurements of the deep venous velocity and flow: To investigate the changes of venous stasis, we divided the participants into the following groups:; 1) non-pregnant women without leg edema,

*Who measures or uses a sampling technique? Please also provide the accuracy, validity, and reliability of the measurement and record it. =>(A) Thank you for good advice. Yes, we inserted the following sentence in Materials and Methods: The same trained evaluator (who had been trained for three months) assessed the measured values under the appropriate conditions.

*Results: please provide the data of the measurement in Table and also provide the statistical analysis.=> We are sorry. We can’t provide all data in Tables. They are huge amount and impractical. Instead, we provided Figure3ABC, Figure 4ABC and Figure 5A1ab,5A2ab, 5B1ab, 5B2ab , 5C1ab and 5C2ab. We provided statistical analyses in these graphs. We think that’s best. We also expressed them in our past papers (Healthcare). We appreciate your understanding.

*Discussion: what is the new knowledge of the study? what is the application of the study. 

=>(A)Thank you very much for your valuable opinion. Yes, we inserted the following sentences in Discussion: These results are new findings and indicate that the risk of DVT formation was higher in pregnant women with leg edema during late pregnancy than without leg edema.

In the present study, the venous velocity and blood flow were significantly increased by wearing elastic compression stockings for leg edema in women in late pregnancy and these phenomena are new discoveries. Thus, pregnant women in late pregnancy with edematous legs should wear elastic compression stockings to prevent DVT, which is the application of this study.

Reviewer 3 Report

Comments and Suggestions for Authors

Dear editor, I would like to thank you for your kind invitation to review the article entitled “The effect of wearing elastic compression stockings on leg edema in pregnant women in late pregnancy as determined by measuring the deep venous velocity and flow.”

The authors aimed to investigate the possibility of preventing DVT in late 78 pregnancy using elastic compression stockings. 

First of all, I would like to congratulate the researchers for their efforts. 

The title and the abstract are adequate. 

The introduction part of the article is sufficient, but  I suggest they to add a hypothesis in this section.

Method section

The type of study was not written. Please add.

Explain the inclusion and exclusion criteria in detail.

The number of patients in the method and abstract differ. these numbers should be checked. Present the number of individuals included in the study in the results section. this belongs to your results.

Please move the table 1 to the result section.

Explain in detail how the socks are worn and for how many hours in a day. How was it assessed whether the patients wore their socks every day?

Review and simplify the repetitions of information in the whole article.

When analysing the power in the study, please indicate for the readers which parameter and which article you refer to.

Results

Please, explain how you calculate the percentage changes.

If it is in accordance with the journal guidelines, I suggest that venous changes should be presented in a table rather than in plain text, as this will allow readers to see the changes more clearly.

Discussion and conclusion

I suggest that the discussion and conclusion section could be improved  further.

I suggest you write in the discussion section what the incidence of DVT in pregnant women is.

I suggest you review the use of punctuation and double spaces throughout the article.

I wondered if the gynaecologist evaluated pregnant women and babies before and after the use of stockings. I suggest that the effects of increased venous return velocity on the baby and the mother's heart should also be discussed.

Comments on the Quality of English Language

The language and expression of the article are appropriate. 

Author Response

We corrected our article with red color (highlight).

<Answers to the referee 3>

Dear editor, I would like to thank you for your kind invitation to review the article entitled “The effect of wearing elastic compression stockings on leg edema in pregnant women in late pregnancy as determined by measuring the deep venous velocity and flow.”

The authors aimed to investigate the possibility of preventing DVT in late 78 pregnancy using elastic compression stockings. 

First of all, I would like to congratulate the researchers for their efforts. 

The title and the abstract are adequate. 

*The introduction part of the article is sufficient, but  I suggest they to add a hypothesis in this section.=>(A) Thank you very much for your good advice. Yes, We inserted the following sentence in Introduction:  We made a hypothesis that venous stasis might worsen during pregnancy, under conditions of leg edema, but improve by wearing elastic stockings.

Method section

*The type of study was not written. Please add.=> (A)Thank you. We add “The study design was non-randomized controlled and intervention trial.” in Materials and Methods.

*Explain the inclusion and exclusion criteria in detail. =>(A) We added the following sentences in Materials and Methods: All non-pregnant and pregnant women didn’t have serious complications. Edematous legs were not pre-eclampsia but physiological. 

*The number of patients in the method and abstract differ. these numbers should be checked. Present the number of individuals included in the study in the results section. this belongs to your results.

=> (A) We checked the sample numbers in Abstract and Materials and Methods. We could not find the different numbers in them. Because it is complicated, we rearranged the following sentence: Research subjects at 36 weeks of gestation were 23 pregnant women (46 legs) without leg edema, 22 pregnant women with leg edema (44 legs) who didn’t wear stocking for 1 week , and 21 pregnant women (42 legs) with leg edema who wore stockings for 1 week.

*Please move the table 1 to the result section.=>(A) Yes, we moved table 1 and the section of Clinical characteristics of the study participants (Table 1) to the Results.

* Explain in detail how the socks are worn and for how many hours in a day. How was it assessed whether the patients wore their socks every day?=>(A) Yes, we inserted the following sentence in Materials and Methods: The pregnant women wore elastic compression stockings from 36 to 37 weeks of gestation every day except for bathing and sleeping.

*Review and simplify the repetitions of information in the whole article.=> Thank you. We deleted the following sentences in Material and Methods:  They were asked not to alter their lifestyle (including water intake) during the study period.

.(non-pregnant women without edema, pregnant women without edema at 36 gestational weeks, pregnant women with leg edema at 36 gestational weeks without treatment for 1 week, and pregnant women with leg edema who wore elastic compression stocking from 36 to 37 gestational weeks.

with leg edema at 36 and 37 gestational weeks, and in the pregnant women with leg edema who wore elastic compression stockings at 36 and 37 gestational weeks

*When analysing the power in the study, please indicate for the readers which parameter and which article you refer to.=> (A)Thank you very much for your nice advice. Yes we added the following sentence in Materials and Methods: Particularly noteworthy is the differences of venous stasis among the 4 study groups. 

Results

*Please, explain how you calculate the percentage changes.=> (A) Yes, we compared A (average±standard deviation) with B (average±standard deviation) and calculated (A average-Baverage/Baverage. Thus, we inserted its numerical formula in Results (first appearance): (6.88 ± 0.31 cm/s vs. 10.17 ± 0.42 cm/s; 32.4% (10.17-6.88/10.17) decrease, p<0.0001).

*If it is in accordance with the journal guidelines, I suggest that venous changes should be presented in a table rather than in plain text, as this will allow readers to see the changes more clearly.=> (A) We are sorry. We expressed their changes in graphs. We also expressed them in our past papers (Healthcare). We appreciate your understanding.

Discussion and conclusion

*I suggest that the discussion and conclusion section could be improved further.=>(A) Thank you for good advice. Yes, we changed Texts in Discussion and Conclusion in accordance with 3 referees’ suggestions.

*I suggest you write in the discussion section what the incidence of DVT in pregnant women is.

=> (A) Thank you for your advice. We inserted the following sentence in Discussion.; Actually, it was reported that the incidence of pregnancy related venous thromboembolism was 1 in 1500 deliveries and the risk of venous thromboembolism was five times higher in a pregnant woman than in a non-pregnant woman [27].

*I suggest you review the use of punctuation and double spaces throughout the article.=>(A) Yes. Editorial office will help us.

*I wondered if the gynaecologist evaluated pregnant women and babies before and after the use of stockings. I suggest that the effects of increased venous return velocity on the baby and the mother's heart should also be discussed.=>(A) Thank you for good idea. Yes, we inserted the following sentences in Discussion.; In addition, the effect of compression stockings on maternal and fetal circulation was studied in left lateral position and standing in 21 patients with uterovascular syndrome in late pregnancy [29]. Thus, the use of compression stockings indicated a measurable improvement in maternal and fetal circulation.

Reviewer 4 Report

Comments and Suggestions for Authors

Compression stockings constitute an important method on noninvasive treatment of edema of lower extremities in pregnant women. This is an interesting study, properly designed and performed.

The introduction provides a background of the presented problem, and methods are clearly described. Results are adequately presented, discussions provides relevant reverences, however some of them should be more recent. conclusions are supported by the results.

Author Response

We corrected our article with red color (highlight).

<Answers to the referee >

*Compression stockings constitute an important method on noninvasive treatment of edema of lower extremities in pregnant women. This is an interesting study, properly designed and performed. =>(A) Thank you very much!

*The introduction provides a background of the presented problem, and methods are clearly described. Results are adequately presented, discussions provides relevant reverences, however some of them should be more recent. conclusions are supported by the results.=>(A)Thank you very much your good advice. New researches regarding stockings effect are few. And we added number 27 reference (2016). Furthermore, we changed conclusion just a little.

Round 2

Reviewer 1 Report

Comments and Suggestions for Authors

The study was presented well. Appreciate for the revised version.

Reviewer 2 Report

Comments and Suggestions for Authors

The revised manuscript could be accepted. 

Reviewer 3 Report

Comments and Suggestions for Authors

Dear editor, the authors have made all the requested revisions. In this sense, I congratulate them and wish them continued success. I think that the article is suitable for publication in this form.

Comments on the Quality of English Language

The English language of the article is appropriate.